# Concentrations of Pb and Other Associated Elements in Soil Dust 15 Years after the Introduction of Unleaded Fuel and the Human Health Implications in Pretoria, South Africa

**DOI:** 10.3390/ijerph191610238

**Published:** 2022-08-18

**Authors:** Joshua O. Olowoyo, Ntebo Lion, Tshoni Unathi, Oluwaseun M. Oladeji

**Affiliations:** Department of Biology and Environmental Sciences, School of Science and Technology, Sefako Makgatho Health Sciences University, Ga-Rankuwa 0208, South Africa

**Keywords:** lead, manganese, iron, vehicular emission, unleaded fuel

## Abstract

Leaded fuel has been reported to contain certain amounts of toxic trace metals such as Pb and Cadmium (Cd), which may have negative impacts on humans and the environment. Unleaded fuel was introduced to South Africa in 2006 with the aim of reducing and eventually eliminating the negative impact of leaded fuel on the environment. However, trace metals are usually nonbiodegradable, and it may therefore be necessary to monitor their presence in the environment so as to evaluate their possible impact on human health. The present study evaluated the levels of Pb and other heavy metals in soil samples collected from petrol (gas) filling stations and from busy roads just around the filling stations in Pretoria, South Africa, fifteen years after the introduction of unleaded fuel. A total of twenty-four (24) soil samples were analysed for lead (Pb), chromium (Cr), copper (Cu), zinc (Zn), arsenic (As), iron (Fe), manganese (Mn), nickel (Ni), titanium (Ti), and cadmium (Cd) using Inductively Coupled Plasma Mass Spectrometry (ICP-MS). The results showed that the concentrations of the trace metals were in the following ranges: Pb, 0.08 ± 0.02–188.36 ± 15.32 ug/g; Mn, 5.35 ± 0.34–6842.43 ± 1.35 ug/g; Zn, 1.82 ± 0.22–9814.89 ± 22.32 ug/g; As, 0.21 ± 0.00–8.42 ± 2.44 ug/g; Cu, 10.51 ± 3.41–859 ± 0.09 ug/g; Cr, 5.80 ± 2.21–417.70 ± 9.08 ug/g; Ti, 19.94 ± 4.99–1036.12 ± 1.49 ug/g; and Fe, 3.06 ± 7.87–674.07 ± 12.22 mg/g. The highest concentrations from all the elements were recorded for Fe in all the collected soil samples. The concentrations of Pb in the soils collected from sites associated with high traffic and industrial areas were higher than for those from all other sites, and the differences were significant (*p* < 0.05). The pollution index (PI), which is the anthropogenic influence of the trace metals, and the geoaccumulation (Igeo), which allows for the removal of possible variations as regards the studied element in the soil due to the possible differences in the background value, showed that some samples were enriched anthropogenically. The PI for Ni, Pb, Cu, and Cr indicated highly anthropogenically contaminated soils, especially at sites associated with high traffic volumes and in industrial areas. The Igeo showed moderately polluted areas for Pb and Cu in high-traffic areas. The exposure routes for the toxic trace metals that were of concern in the study were either through ingestion or dermal contact. The calculated hazard quotient showed both noncarcinogenic and carcinogenic risks for Fe and Mn via ingestion and through dermal contact for both children and adults, respectively. The concentrations of Pb were high and similar to those that were previously reported in the study and pointed to vehicular emission as one of the contributors. The study also noticed an increase in the presence of Mn and Fe in all soil samples.

## 1. Introduction

Trace metal pollution in the environment is a worldwide problem because trace metals are nonbiodegradable and most have hazardous effects on living creatures when permissible concentration levels are surpassed. Cadmium (Cd), chromium (Cr), and lead (Pb) are known to be toxic trace metals, especially if the levels are above the recommended limit. Zinc (Zn), iron (Fe), and copper (Cu) are trace elements that are commonly documented in the literature as macroelements, though higher levels in polluted soils may be associated with health risks [1]. Soil pollution from trace metals plays a significant role in both metal geochemical cycling and environmental health. Increased levels of trace metals in the soil have raised serious concerns regarding their impact on plant and animal life. While some of these trace metals, such as Cu, Zn, Mn, and Fe, are necessary nutrients for enzymatic and biochemical activity in the body, others, such as Cd, Pb, As, and Hg, may be toxic even at low concentrations—especially in cases of continuous exposure [2].

The historical usage of Pb additives in gasoline has contributed significantly to the majority of anthropogenic Pb in the urban environment; however, this might have decreased since the ban went into place in 2005 [3,4]. Pb is a naturally occurring element that has been employed in a variety of applications throughout the twentieth century, including gasoline, paint, and plumbing [5]. Anthropogenic Pb produced from gasoline applications has been identified as an environmental pollutant all over the world. Unfortunately, because Pb can be toxic, it may have negative impacts on human health. However, due to the nonbiodegradable nature of Pb, its fingerprints endure in both urban and distant soils, posing a long-term public health concern [6].

In South Africa, leaded gasoline was banned in 2006 and was replaced by unleaded gasoline. The phase-out of leaded gasoline was expected to not only reduce the introduction of Pb into the environment, but to also support the realization of various sustainable developmental goals such as good health and well-being, clean water, clean energy, sustainable cities, sustainable climate action, and sustainable life on land. This action was also expected to assist in creating a pathway for the restoration of ecosystems that have been damaged by various harmful contaminants, including the atmospheric Pb emitted by leaded fuel [7]. It should also be noted that unleaded gasoline contains 15 dangerous compounds, including trace metals, benzene, toluene, naphthalene, and trimethyl benzene, among others [8].

In some studies, unleaded gasoline has already been identified as a new anthropogenic source of Pb dispersion into the atmosphere [9,10]. According to Shiel et al. [9], unleaded gasoline usage accounts for 8.4 percent of the total Pb emissions from fossil fuels [11]. However, reports from the literature have revealed some of the repercussions of these changes for humans. For example, Batterman et al. [12] discovered a trend in the form of a dramatic reduction of Pb concentrations in both the air and children’s blood in Durban, South Africa. While some children exceeded the guideline levels for Pb, as directed by [13], levels of Mn in the blood of the children who participated in the study were within the acceptable limit—though higher values were reported with children of Indian descent. The study also linked the sources of Pb to sources other than automobile pollution [12]. Rollin et al. [14] also reported the beneficial effects of unleaded gasoline on Pb in utero levels in South African populations; the study found that the adoption of unleaded gasoline and lead-free paint had a positive influence. Apart from the gasoline, toxic trace metals could be derived from several other anthropogenic sources in the urban environment, including industrial operations, traffic emissions (vehicle exhausts and wear products from tires, brake linings, and bearings), and natural sources [15].

Despite the detrimental effect of gasoline in the environment, its usage in South Africa has not decreased due to the increased demand as a result of industrialization and urbanization [16]. This increased demand has prompted the establishment of several gas stations. Furthermore, the country’s current inconsistent supply of electricity has resulted in a rise in the use of power generators to produce electricity, which also requires the usage of gasoline [17], which may also increase the introduction of toxic trace metals from the gasoline into the environment.

Although unleaded petrol has been available in South Africa since 2006, there are few studies exploring the impact of the introduction of unleaded gasoline to soils in South Africa—in particular, there are few studies using Pb and associated trace metal levels to assess the impact of unleaded fuel. Olowoyo et al. [16] discovered a significant concentration of trace metals such as Pb, Mn, As, and Cd in soils from the urban and residential areas of Pretoria six years after the introduction of unleaded fuel. Due to the amounts reported in the soil samples during the study, the study concluded that Pb might not be the only trace metal polluting the environment, and that Mn could also be a significant pollutant. Pb has been reported to be extremely harmful to humans and other animals around the world, as long-term exposure can lead to bioaccumulation and biomagnification, resulting in serious neurological health problems [18,19]. The gastrointestinal tract, kidneys, and central nervous system are frequently affected by lead poisoning. Lead-exposed infants are more likely to have delayed development, a lower IQ, a shorter attention span, hyperactivity, and mental problems [20]. When adults are exposed to Pb, they typically experience slower reaction times, memory loss, nausea, insomnia, anorexia, and joint weakness. Lead has no known physiological function in the human body; it can only cause harm when ingested through air, food, or water. Mn is also known to affect the nervous systems, leading to behavioural changes and, in some instances, may also lead to a loss in sex drive in adult males, with damages to the sperm [21]. Other investigations on the amounts of Pb and Mn in Pretoria street dust, published previously by Okonkwo et al. [22], revealed significant concentrations of Mn and Pb in urban soils as compared to soils from rural areas, with concentrations ranging from 279 ug/g to 864 ug/g, respectively, and at amounts higher than the acceptable limits recommended by [23]. The overall concentration of Pb in the soils may not provide precise information on whether the source of Pb as a pollutant emanates only from leaded fuel, so other isotopes of Pb—that do not undergo fractionation—may provide a clearer picture on the source of Pb in the environment [24]. The current study investigated the concentrations of Pb in soil samples that were collected from different areas in Pretoria with a view to investigate the impact of unleaded petrol on reducing the concentrations of Pb in soils 15 years after the introduction of unleaded fuel in South Africa. This will be compared with previous studies carried out in the area. The goal is to evaluate the relationship between the use of unleaded fuels and levels of Pb and other trace metals in soil. The study will also highlight any potential human health risks that might be linked to the presence of Pb and other trace metals in soil.

## 2. Materials and Methods

### 2.1. Study Area

The study was carried out in Pretoria, one of the three capital cities of South Africa. The city is the seat of the executive branch of government, also housing many foreign embassies in South Africa. Currently there are about 2,655,000 people in the city (STAT SA 2019). The climate is humid, accompanied by long, hot, rainy summers, with short but mild winters. The city experiences winter during the months of May–August, followed by spring in September (but hot), while the summer season is officially experienced from December–April. The average annual temperature is 18.7 °C. The city has a relatively high altitude of about 1339 m. Rain is chiefly concentrated in the summer months, with drought conditions prevailing over the winter months with no snowfall.

### 2.2. Sampling

The soil dust samples were collected within and around petrol stations (gas stations) in the city. Eight sampling stations were used in the study, and these comprise the main city (Pretoria central, industrial area, suburbs of the town, as can be seen in Figure 1). From each sampling site, three samples were collected; each sample consisted of 5 subsamples which were homogenized to one sample. The petrol stations and the surrounding areas were chosen for this study because these were considered as the main sources of unleaded fuel, and because vehicles are expected to queue up in this area on a daily basis. The three locations where soil samples were collected from each of the sites were at the point of entry (the main road leading to the gas station, selected based on close proximity to main roads and movements of the vehicles), the midpoint inside the petrol stations (vehicles usually queue up in these areas before refilling their fuel; engines may be switched off while in the queue), and areas just beside the petrol pump. A clean plastic parker and brush were used for soil sample collections. After each collection, the plastic parker and brush were thoroughly dusted and cleaned so as to avoid cross contamination. The original weight of each sample was maintained at 2 kg to ensure that the weight after drying and sieving was not less than 500 g [25].

### 2.3. Sample Preparation and Analysis

In the laboratory, soil dust samples were air dried at room temperature. Debris, pebbles, and rocks were removed from the samples. Each sample was then sieved through a mesh of 100 um in size for the determination of trace metals. The chemical analyses were conducted following the method described by [26] and in accordance with standard practice for trace metal determinations in soil by the South African Agricultural Council. A complete digestion procedure was followed, wherein 0.5 g of the soil sample was weighed and 8 ml HNO_3_, 2 mL of HClO_4_, 2 mL of HF, and 2 mL HCl were added for the purpose of digestion. After heating up for 2 h, the resulting solution was made to volume in a 50 mL flask by adding distilled water [26]. The heavy metal content in the soil samples was determined using ICP-MS (ICP-MS 7900, Thermo Fisher, Waltham, MA, USA). All the samples were completed in triplicates. The quality assurance and quality control—standard procedures described by the South African Agricultural Council—were followed as well, and the certified reference material (Light alluvial–deluvial meadow soil 0001-1999-BG) was bought from China’s National Center for Standard Reference Materials. The blank corrections were also analysed concurrently to ensure that the results were accurate, with a standard deviation of less than 3%. The recoveries for all trace metals (Cd, Pb, As, Zn, Cu, and Mn) ranged from 95% to 100%.

### 2.4. Pollution Assessment

The influence of anthropogenic activities on heavy metal pollution levels in filling station soil were investigated using geoaccumulation index (Igeo) and pollution index (PI) [27,28].

#### 2.4.1. Geoaccumulation Index (I_geo_)

The geoaccumulation index (I_geo_) was calculated using the method modified by [29]. The purpose of calculating the geoaccumulation index was to eliminate human interference while assessing the levels of contamination and this is calculated in relation to the background values.
I_geo_ = log_2_ C_n_/1.5B_n_(1)
where C_n_ is the measured concentration of trace metal (n) in the soil samples. B_n_ is the geochemical background values of element (n) in soils [26]. The background values are: As, 2 ppm; Cd, 3 ppm; Cr, 350 ppm; Cu, 120 ppm; Ni, 150 ppm; Pb, 100 ppm; and Zn, 200 ppm [30].

1.5 is the background matrix correction factor [30].

Result interpretations:

I_geo_ < 0 designates uncontaminated,

0 ≤ I_geo_ < 1 designates uncontaminated to moderately contaminated,

1 ≤ I_geo_ < 2 designates moderately contaminated,

2 ≤ I_geo_< 3 designates moderately to strongly contaminated,

3 ≤ I_geo_ < 4 designates strongly contaminated,

4 ≤ I_geo_< 5 designates strongly to extremely contaminated,

I_geo_ ≥ 5 indicates extremely contaminated.

#### 2.4.2. Pollution Index (PI)

The pollution index (PI) was used to determine the magnitude of soil contamination in relation to trace metals in the soils. The pollution index (PI) is simply defined as the ratio of the concentration of trace metals to the background value of the corresponding metal [26].

It is usually determined by this relation:Pl = C_i_/S_i_(2)
where C_i_ is the recorded value of each metal (mg/kg) from the soil samples and S_i_ is the background value (mg/kg) [30].

When the calculated PI is greater than 1, it indicates the influence of anthropogenic factors.

### 2.5. Human Health Risk Assessment

Human health risk assessment entails evaluating potential human health effects in contaminated environmental media [31]. The human health risk assessment was used to assess the carcinogenic and noncarcinogenic risks associated with dermal contact and ingestion exposure pathways [32].

#### 2.5.1. Exposure Assessment

The health risk assessment is based on the United States Environmental Protection Agency’s exposure factors and guidelines handbook [33]. The average daily dose (ADD) for both children and adults was evaluated via inhalation (ADD_inh_), ingestion (ADD_ing_), and dermal contact (ADD_dermal_) following Equations (1) and (3):(3)$$ADD_{ing}=\frac{C_{Siol}*IngR*EF}{BW*AT*10^{6}}$$
(4)$$ADD_{inh}=\frac{C_{Siol}*InhR*EF*ED}{PEF*BW*AT}$$
(5)$$ADD_{dermal}=\frac{C_{Siol}*SA*AF*ABF*EF*ED}{BW*AT*10^{-6}}$$

The risk assessment was conducted using the following exposure factors and values seen in Table 1 [28].

#### 2.5.2. Noncarcinogenic Risk Assessment

The hazard quotient (HQ), carcinogenic risk (RI), and hazard index (HI) approaches were used to assess carcinogenic and noncarcinogenic adverse effects [28]. The hazard quotient [34] is defined as the average daily dose of heavy metals divided by the reference dose (RFD):(6)$$HQ=\frac{ADD}{RFD}\;$$

If HQ < 1, it means there will be no negative effects; if HQ > 1, there will be negative effects [35].

The hazard index approach was used to evaluate the noncarcinogenic risk’s overall negative effects [36]. The HI is the total of the HQ for the three heavy metal/metalloid exposure pathways. According to [34], the HI was assessed for a combination of contaminations:(7)HI=∑HQ

While HI > 1 indicates negative effects, HI < 1 indicates noncarcinogenic effects.

#### 2.5.3. Carcinogenic Risk Assessment

The carcinogenic risk is the possibility of developing cancer throughout an individual’s lifetime exposure to carcinogenic risks [31,37]. According to [34,38], the slope factor (SF) directly converts the ADD of a pollutant exposed over a cancer patient’s lifetime risk. Table 1 and Table 2 displays the values for SF, RFD, and other derived parameters.
(8)Risk=ADD∗SF

A risk number of 10^−6^ indicates that there is no carcinogenic risk to health from the dust, whereas a risk value of >1 × 10^−4^ indicates a high chance of acquiring cancer. A risk number between 1 × 10^−6^ and 1 × 10^−4^ indicates an acceptable risk to human health [37].

### 2.6. Statistical Analysis

The statistical analysis was performed using the SPSS 28.0. Analysis of variance was used to determine if the differences obtained in the concentrations of trace metals from the different sites used in the study were significant. Pearson correlation was also used to determine if there is a relationship between the trace metals recorded from all the sites.

## 3. Results and Discussion

The concentration of trace metals (Cr, Mn, Fe, Ni, Cu, Zn, As, Cd, and Pb) in the soil samples collected from the sites used for this study (Pretoria suburb, industrial, city centre, and residential areas) are presented in Table 3.

Description: P1 is the first point where samples were collected, P2 represents the second area, and P3 represents the third area. S1, S2, S3, …, S8 are the locations. The suburbs are two different localities which are not located far from the Pretoria city centre. Suburb 1 is 40 km away from the Pretoria city centre, while Suburb 2 is 37 km away from the Pretoria city centre. The residential area is 24 km to the Pretoria city centre. Generally, the values reported for all the elements from the Pretoria city centre locations (city centres 1 and 2) were the highest when compared with all other sites. This was followed by the reports obtained for the two industrial areas used in this study. The differences obtained for these results from all the sites were significant (*p* < 0.05). The high concentrations of trace metals in the Pretoria city centre could be related to vehicular emissions releasing trace metals into the environment. A similar finding was noticed in the study by [39], wherein it was reported that the source of the trace metals in the study might have emanated from atmospheric deposition originating from the combustion of fossil fuels and gasoline coming from vehicular emissions.

The results in Table 3 showed the presence of the analysed trace metals in the samples, which reflects a widespread and diffuse pollution of trace metals in the soil of the study area. The overall results showed: As (0.09 ± 0.02 µg/g–8.42 ± 2.44 µg/g); Pb (1.16 ± 0.41 µg/g–188.34 ± 3.98 µg/g); Cd (0.06 ± 0.01 µg/g–1.59 ± 0.09 µg/g); Cr (5.80 ± 2.21 µg/g–417.70 ± 9.08 µg/g); Ni (8.91 ± 3.22 µg/g–486.60 ± 6.11 µg/g); Zn (58.30 ± 4.21 µg/g–1675.21 ± 7.76 µg/g); Mn (144.14 ± 1.11 µg/g–6842.43 ± 7.88 µg/g); Fe (3.06 ± 1.78 mg/g–674.07 ± 12.22 mg/g); and Cu (10.51 ± 3.41 µg/g−859.48 ± 0.09 µg/g). The trend was in the order Fe > Mn > Zn >Ni > Cr > Cu > Pb > As > Cd, respectively. There were positive correlations between some of the trace metals recorded in the study—for instance, between Pb and Cd (0.85), Pb and Zn (0.87), Pb and Ni (0.60), Pb and Mn (0.60), Ni and Mn (0.90), and Mn and Zn (0.83).

Trace element concentrations, particularly Fe, Mn, and Zn, were all higher than the maximum levels allowable in soils [23]. This indicates that these three heavy metals are the most prevalent pollutants in the soils of the sites owing to their concentrations that were above the acceptable limit. This is in line with the study by [30], in which it was previously reported that these three elements exhibited values above the baseline reported in soils from South Africa. From this current study, the presence of these pollutants in the soil of the study area may be partly due to exhaust from automobiles and gasoline combustion. Similar studies from [40,41] showed that the source of these trace elements could also be traced to industrial activities, and both studies attributed them to mining industries. It has been reported that the combustion of gasoline and the emissions from automobiles are the main sources of these elements in air pollution (particularly lead particles), reaching soils via dry and wet depositions [42,43]. According to some studies, motor vehicles are the primary source of these metals [44,45].

From Table 4, the maximum concentrations recorded for Pb, Cr, Ni, Mn, and Fe in some of the sites used in this study were all above the maximum limits set by the [23]. The highest concentration recorded for Cr was four times higher than the recommended limit. The highest concentrations for Ni were 10 times higher than the maximum recommended limit—similar to the highest value recorded for Cu. This clearly shows that some of the levels of toxic trace metals in the soil may be toxic owing to their high concentrations in the soil. This could be especially dangerous in a country where pregnant women practice geophagia [46].

The concentrations of Pb in this study showed a similar trend to that which was previously reported by both [16,22], wherein areas associated with high traffic volumes exhibited the highest concentrations of Pb. From the current study, the highest concentrations of Pb were all recorded from the Pretoria city centre areas, which happen to be in a busy centre with bubbling commercial activities and a high volume of vehicles. These areas were previously designated as high-traffic areas with resulting high concentrations of Pb. The concentrations recorded for Pb from these areas were two- or three-fold more than those recorded from the other sites with less traffic—especially when compared to sites from the suburb—and the differences obtained were significant (*p* < 0.05). The previous values reported by [16] showed that the values of Pb were in the range 12.8–145 µg/g. This revealed that there were no significant changes on the levels of this toxic trace metal in the soil. As a result, the source of Pb in these areas may not be entirely due to anthropogenic sources or from vehicular emission but may be a result of the nonbiodegradable nature of the toxic trace metal and resuspension in the environment [42,43]. It must also be noted that some studies have pointed out that unleaded gasoline may not entirely eradicate the presence of Pb in the environment because unleaded gasoline has been reported to contain a low amount of Pb [47]. In this study, the trend is similar and seems to be more noticeable in areas associated with vehicular emissions. The Pb levels in the study area could be linked to automotive tailpipes, which account for nearly two-thirds of Pb emissions into the atmosphere [48]. The findings of these studies conforms with the assertion made from some studies that were conducted recently, which clearly shows that even decades after the Pb ban, historical Pb stored in soils may serve as a persisting source of Pb in the environment due to the remobilization and deposition of contaminated dust because it is nonbiodegradable [6,49].

Among the heavy metals, lead (Pb) is the most well-known immobile nonessential element, with the majority of Pb-based compounds being toxic in nature. Pb is present in the earth’s crust at a concentration of 0.1 mg/kg on average. Because Pb is a metal toxicant, it is gradually being phased out of materials commonly used by humans. Pb enters the human or animal metabolism primarily through the food chain. When compared to studies from other countries, the observed Pb content in the samples were comparable [23,50].

Similar trends noticed for Pb were also observed for Zn in this study. The concentrations of Zn were higher at both the Pretoria city centre and industrial areas. A positive correlation of 0.85 was reported for Pb and Zn, suggesting a common source. Zn is an important trace element in biological systems due to its enzymatic and regulatory properties. The high concentrations of Zn noted in areas associated with traffic may be due to oil spills from subsurface storage tanks or from the wear and tear of tires [51]. Zn is also one of the heavy metals that can be released during abrasion and wear, as it is a component of tires and motors. High concentrations of zinc (Zn), a readily mobile element, can cause serious haematological and neurologic complications, liver and kidney disorders, hypertension, gastrointestinal misery, loose bowels, pancreatic harm, and a variety of other ailments in both humans and animals [52]. Zinc is found in the earth’s crust at an average concentration of 80 mg/kg in association with other metal ores such as Cu, Pb, and Cd [22,50,53].

The concentrations of Mn in the study were found to be relatively high compared to previous studies reported by [16,22] from the study sites and may be considered as an emerging pollutant in the area due to the current concentrations, especially in areas associated with high vehicular movements which are the city centres 1 and 2 and the industrial areas (Table 2). From these areas, the Mn concentrations were slightly above the acceptable specified limits [23] at some sites. As a result, Mn might be said to be derived not only from parent material in the soil but also from anthropogenic sources. Mn concentrations in the soil samples clearly demonstrate that its presence may be related to automotive usage because of its use in gasoline. Mn is also one of the heavy metals that can be released during abrasion and wear, as it is a component of tires and motors. The study by [54] linked traffic densities to the release of Mn in urban soils and may come from car tires, lubricants, and brake abrasion.

The average Cu concentration of 180.481 μg/g was above the Republic of South Africa’s maximum acceptable concentration safe limit of 6.6 mg/kg for agricultural soil [23,55]. Cu is an essential micronutrient for the growth of both plants and animals. It aids in the production of blood haemoglobin in humans, and it is used by plants for seed production, disease resistance, and water regulation. Cu can cause anaemia, liver damage, kidney damage, and stomach and intestinal irritation at high doses [56]. Copper (Cu) may also be considered a pollutant in this study due to its levels from all the sites, but it followed a similar trend of having high concentrations in areas associated with high vehicular movements and industrial sites, which may also suggest an external influence.

Cadmium (Cd) concentrations in soil samples from some sites were below the detection limit. The highest concentration was recorded from a soil sample collected from the industrial area with a mean value of 1.59 ± 0.09 μg/g. This was closely followed by soil samples collected from the city centre, which had a mean value of 1.51 ± 0.04 ug/g. A trend similar to the ones reported for Pb, Mn, and Cu was also noticed for Cd. The concentrations reported for Cd in this study were all below the acceptable limit of 1–3 mg/kg in soils, as recommended by the European Union. Cadmium (Cd) has been identified as one of the most environmentally toxic metals, with the potential to harm biological activities, plant metabolism, soil health, and human health [57]. In Ni/Cd batteries, Cd is widely used as a rechargeable or secondary power source with high output, long life, low maintenance, and high tolerance to physical and electrical stress. Because Cd is very biopersistent and, once absorbed by an organism, remains resident for many years, observed levels are of great concern. Cd has been shown to affect several enzymes in humans. According to previous research, Cd adversely affects enzymes responsible for protein reabsorption in kidney tubules, resulting in renal damage and proteinuria [58]. Long-term exposure to this metal lowers the activity of alcohodehydrogenase, delta-aminolaevulinic acid synthetase, arylsulfatase, and lipoamide dehydrogenase, all of which cause cardiovascular disorders, anaemia, and hypertension, while increasing the activity of delta-aminolaevulinic acid dehydratase, pyruvate dehydrogenase, and lipo [53].

In the current study, the levels of As and Cd in soil samples were low. As and Cd do not appear to play a significant role in the contamination of the areas under investigation according to these findings. The concentrations of contaminants in soil at the point of exposure may differ from the concentration at the source due to fate and the transport processes (e.g., dispersion, biodegradation). However, despite the relatively low values for these trace metals, it is always advisable to determine the exposure to contaminants in soil and dust so as to determine the overall effect on human health.

### 3.1. Pollution Index

The pollution index measured the amounts of heavy metals or metalloids present in the study area in comparison to the background level of those same substances [59]. The study recorded pollution index values greater than one from most of the areas within the main city centres and industrial areas [60]. This suggests that anthropogenic contamination from various activities in these areas might have contributed to the pollution levels of these trace metals. The pollution index was greater than one from city centre and industrial areas for Cr, Cu, Zn, Pb, and Ni with the exception of As. Previous studies have reported on the increase in the levels of these elements in soils from urban areas. Wang et al. [61] noted that soils in urban environments may become a sink to metal (loids) pollution due to the potential impact of fossil fuels and including their widespread use in building materials as a result of developmental projects. In Figure 2, it clearly shows that the average pollution index for Cu was the highest, followed by Zn and Ni. It is evident from Table 5 that the city centre used in this study contributed greatly to the levels of Cu in the study area.

### 3.2. Index of Geoaccumulation

Table 6 shows the index of geoaccumulation of the trace elements from the soil samples from the various study sites. The geoaccumulation index is used to quantify the degree of anthropogenic contamination and compare different metals that appear in different ranges of concentration in the sludge. In this study, the background values used by [55] for South African soils were used. It should be noted that South Africa has rich mineral resources and, similar to other countries, may face a deterioration in soil quality [26]. As seen in Table 6, the levels of pollution were low for the majority of the trace metals studied, except for in a few instances wherein moderately contaminated levels were recorded—these were mostly in the urban city centre and in the industrial areas for Zn and Pb. However, the result of this study should be interpreted along with the background values used in this study and the nature of the soil from the country. For instance, Pb (0.15) and (0.33), from the city centres sites S6P1 and S8P1, were moderately contaminated.

### 3.3. Human Health Risk Assessment

#### Exposure Assessment

Table 7 shows the calculated mean daily heavy metal exposure for adults and children from the soil samples via inhalation, dermal contact, and ingestion. Maximum values obtained for each of the trace elements were used in the study. From Table 7, the following were the ADD trends for heavy metals via ingestion, inhalation, and dermal contact: Fe > Mn > Zn > Ni > Cr > Pb > As, Cd both for adults and children. Ingestion was the most dangerous method of exposure for adults and children, followed by inhalation and dermal contact was the least dangerous. Geophagia is common in the country and is practiced by both adults and children. As such, this may pose serious health risk because most of the soils consumed are usually contaminated with trace metals [62]. The trend in the exposure risk from this study is similar to that describe by [32].

Table 8 shows the results of the health risk assessment. The calculations on the health risks posed to adults and children (Table 8) were adopted from [63]. Hazard quotients (HQ)/HI greater than or equal to 1, and cancer risk (CR) from 10^−4^ to 10^−6^ were considered significant in this study. From this study, the HI values for soils for the adult and children via ingestion had values greater than one, which indicated a serious threat if the soils collected from the most polluted area—which in this case is the urban city centre—are consumed. Dermal contact for the adults seems to be another dangerous route. This indicates that more attention should be paid to the city centre where there are higher levels of these pollutants compared to other sites.

The average routes of heavy metals exposure for adults and children were in the following order: inhalation < dermal contact < ingestion. In this investigation, ingestion and cutaneous (dermal) exposure were found to be dangerous. This study’s findings are consistent with earlier studies reported by [64], which similarly revealed that ingestion was the most common route to health risk. Except for ingestion in minors, the HI findings from inhalation and dermal contact for all the heavy metals were less than 1, indicating no noncarcinogenic danger. This is due to the fact that children have extensive contact with soil during outdoor play activities and are more likely to have direct hand-to-mouth soil exposure [65]. Other studies with similar findings include [1,66,67].

The RI values in the study area were 1.01 × 10^21^ (As), 1.01 × 10^19^ (Pb), 8.80 × 10^18^ (Cd), and 1.75 × 10^−4^ (Cr) for adults and 2.58 × 10^−5^ (As), 2.57 × 10^−6^ (Pb), 2.24 × 10^−6^ (Cd), and 4.51 × 10^−4^ (Cr) for children (Table 9). The risk indices for Pb and Cd were 10^−6^ for children, and 10^19^ and 10^18^, respectively, for adults, indicating no conceivable carcinogenic risk. However, the HI values in Table 7 for ingestion, on the other hand, imply carcinogenic dangers from the identified heavy metals and this cannot be neglected.

## 4. Conclusions

The concentrations of Pb and other trace elements were determined in the soils of Pretoria 15 years after the ban of leaded fuel in South Africa and the subsequent introduction of unleaded fuel, with a view to establish the impact of unleaded fuel on the levels of Pb and other trace elements. This is the first study to evaluate the impact of the unleaded fuel on the levels of trace metals, making particular reference to Pb after the ban of leaded fuel in Pretoria. From the study, the concentrations of Pb in the soil samples were clearly associated with traffic volumes. Areas with high traffic recorded the highest concentrations of Pb in the study and this may be due to resuspension as a result of the nonbiodegradable nature of the toxic trace metals. A positive correlation was observed between Pb and some trace elements such as Cu, Zn, Ni, and Cd. However, the levels of Fe and Mn in this study were significantly higher than the recommended limits from either the WHO/FAO. This may be of particularly serious concern when the levels of Mn are considered. Previously, the levels of Fe were high, but an appreciable increase in the levels of Mn was reported in this study. This may be a cause for concern owing to the detrimental effects of Mn on human health. The concentrations of elements varied significantly across all sampling sites (*p* < 0.05). The PI and Igeo revealed the possibility of anthropogenic sources for these trace metals in the environment. The PI values greater than one that were recorded for Ni, Pb, Zn, and Cu in some of the soils suggested a moderate-to-high level of pollution, which could be attributed to vehicular emissions, resuspension over time, lubricating oil, bearing wear, and tire wear. The exposure route that might be of concern in this study is through ingestion and followed by dermal contact for both adults and children. This is of concern because of the practice of geophagia, which is common in the country and is practiced by both adults and children. The calculated HI index showed that the areas associated with high levels of contaminants may pose a significant health risk for both children and adults owing to the values greater than one that were reported from these sites. For both children and adults, the most common route of exposure was ingestion, followed by skin contact and inhalation. Constant monitoring of the soils in the urban city centre is recommended in order to understand the nature and levels of these toxic trace metals that may act as pollutants in the city.

## Figures and Tables

**Figure 1 ijerph-19-10238-f001:**
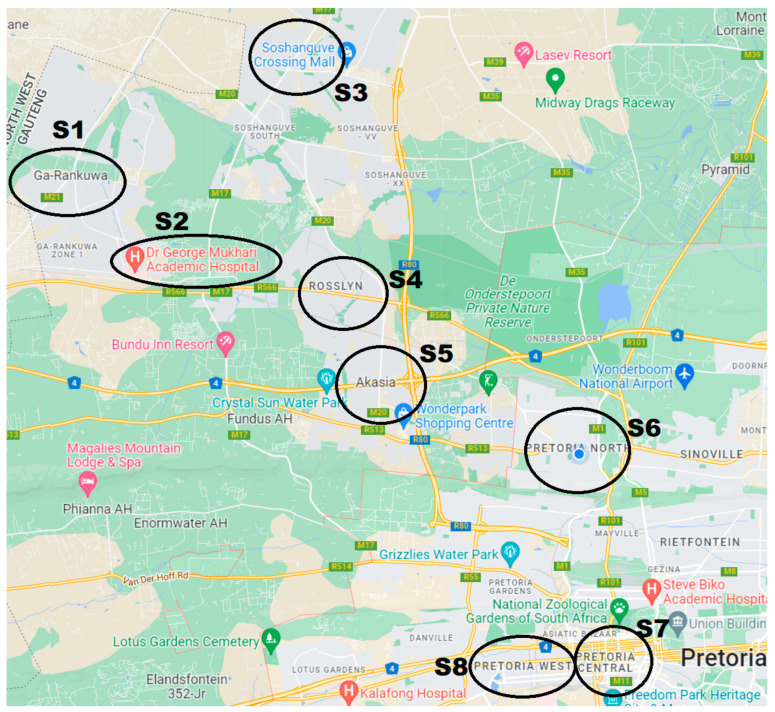
Showing the map of the study area in Pretoria, South Africa.

**Figure 2 ijerph-19-10238-f002:**
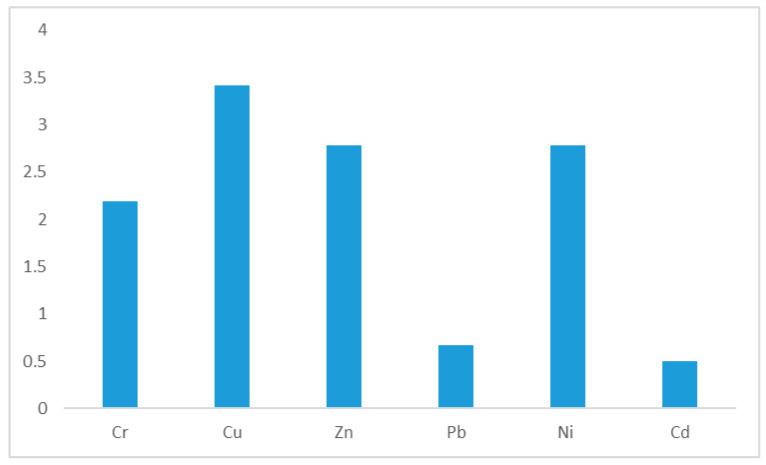
Representation of the pollution index values of the study sites.

**Table 1 ijerph-19-10238-t001:** Showing the exposure values.

Metals	Oral RFD (mg/kg/day)	Oral SF (mg/kg/day)
As	3.0 × 10^−4^	1.5
Pb	3.5 × 10^−3^	8.5 × 10^−3^
Cd	1.00 × 10^−3^	0.5
Cr	3.0 × 10^−3^	0.5
Ni	2.0 × 10^−2^	n.d
Mn	1.4 × 10^−1^	n.d
Fe	7.0 × 10^−1^	n.d
Cu	4.0 × 10^−2^	n.d

**Table 2 ijerph-19-10238-t002:** Represent the risk assessment value.

	Adult/Children
Ingestion rate (IngR)	200/100 (mg/day)
Exposure frequency (EF)	350/350 (days/year)
Exposure duration (ED)	24/6 (years)
Body weight (BW)	55.9/15 (Kg)
Average time (AT)	365 days × ED
Inhalation rate (InhR)	12.8/7.63 (m^3^/day)
Particle emission factor (PEF)	1.36 × 10^9^
Exposure skin surface area (SA)	4350/1600 (cm^2^)
Skin adherence factor (AF)	0.7/0.2 (mg/cm)
Dermal absorption factor (ABF)	0.001

**Table 3 ijerph-19-10238-t003:** Concentrations of trace elements (µg/g and mg/g *) in soils collected from different areas in Pretoria.

		Trace Metals									
Sites	Area	Ti	Cr	Mn	Fe *	Ni	Cu	Zn	As	Cd	Pb
S1P1	**Suburb 1**	53.19± 1.23	11.80 ± 2.11	201.70 ± 1.11	5.69 ± 2.32	12.65 ± 2.89	16.80 ± 3.44	58.30 ± 4.21	0.29 ± 0.02	0.06 ± 0.01	2.25 ± 0.98
S1P2		30.28± 1.22	8.36 ± 3.42	164.38 ± 7.21	3.06 ± 7.87	9.42 ± 4.32	15.61 ± 8.11	70.92 ± 0.98	0.27 ± 0.01	0.21 ± 0.03	3.70 ± 0.20
S1P3		19.94 ± 4.99	5.80 ± 2.21	144.14 ± 1.11	3.29 ± 2.22	8.91 ± 3.22	10.51 ± 3.41	40.56 ± 6.40	0.21 ± 0.00	0.08 ± 0.01	1.16 ± 0.41
S2P1	**Suburb 2**	358.35 ± 7.23	75.85 ± 8.99	1298.08 ± 0.88	38.72 ± 7.43	71.77 ± 0.89	94.48 ± 4.90	317.08 ± 5.64	1.79 ± 0.56	0.33 ± 0.08	20.37 ± 1.21
S2P2		539.45 ± 2.32	93.45 ± 5.65	1144.09 ± 4.76	42.48 ± 2.44	55.86 ± 5.45	86.41 ± 3.43	287.59 ± 0.98	2.58 ± 1.11	0.27 ± 0.03	17.69 ± 0.74
S2P3		43.78 ± 2.09	11.44 ± 1.43	182.52 ± 0.87	5.46 ± 0.09	10.31 ± 0.88	39.92 ± 2.11	81.31 ± 3.11	0.34 ± 0.06	0.10 ± 0.01	3.98 ± 0.08
S3P1	**Suburb 3**	129.95 ± 0.98	103.30 ± 0.11	1891.96 ± 0.99	25.12 ± 0.09	102.83 ± 0.05	219.35 ± 0.98	748.83 ± 0.43	1.08 ± 0.04	0.78 ± 0.22	62.56 ± 0.56
S3P2		566.30 ± 7.68	201.06 ± 0.11	3723.55 ± 2.67	117.27 ± 17.80	189.06 ± 0.09	224.15 ± 0.88	591.15 ± 1.11	4.58 ± 0.98	0.81 ± 0.03	38.55 ± 2.51
S3P3		1036.12 ± 1.49	362.12 ± 12.10	6727.55 ± 1.34	208.49 ± 1.87	326.24 ± 2.44	377.79 ± 2.54	940.16 ± 1.11	8.12 ± 0.01	1.31 ± 0.03	66.65 ± 9.76
S4P1	**Industrial 1**	248.46 ± 2.12	182.20 ± 2.65	4302.74 ± 2.76	124.12 ± 7.66	227.16 ± 4.11	276.55 ± 1.11	818.57 ± 1.98	4.02 ± 2.09	0.97 ± 0.61	114.58 ± 5.89
S4P2		277.99 ± 0.07	105.66 ± 2.44	2141.93 ± 7.98	63.80 ± 8.99	108.76 ± 9.08	125.62 ± 9.11	404.58 ± 7.87	2.31 ± 0.81	0.42 ± 0.01	22.08 ± 0.09
S4P3		405.41 ± 4.22	402.90 ± 12.11	9814.90 ± 7.88	228.72 ± 7.99	486.60 ± 6.11	398.36 ± 7.88	1169.38 ± 4.06	6.84 ± 0.89	1.59 ± 0.09	74.28 ± 9.08
S5P1	**Industrial 2**	219.78 ± 2.33	77.43 ± 8.77	1288.81 ± 2.98	38.97 ± 7.88	68.24 ± 7.88	149.16 ± 78	516.63 ± 89	1.13 ± 0.08	0.49 ± 0.03	26.41 ± 1.15
S5P2		143.98 ± 0.98	46.02 ± 9.07	805.38 ± 0.09	22.85 ± 9.91	46.36± 5.77	102.72 ± 8.99	313.05 ± 6.77	0.71 ± 0.01	0.26 ± 0.01	12.57 ± 0.08
S5P3		122.11 ± 3.11	34.84 ± 7.44	708.23 ± 0.06	18.98 ± 2.34	34.44 ± 3.44	89.90 ± 9.88	192.09 ± 3.34	0.09 ± 0.02	0.98 ± 0.02	15.89 ± 1.12
S6P1	**City Centre 1**	655.41 ± 1.21	243.79 ± 0.98	4549.36 ± 9.12	138.97 ± 5.45	224.71 ± 8.97	404.40 ± 9.12	1675.21 ± 7.76	3.72 ± 1.11	1.41 ± 0.22	188.34 ± 3.98
S6P2		800.82 ± 4.55	298.52 ± 9.11	6842.43 ± 7.88	213.82 ± 9.99	334.20 ± 7.11	329.54 ± 7.99	980.98 ± 9.08	8.42 ± 2.44	1.12 ± 0.09	55.00 ± 1.21
S6P3		908.64 ± 7.99	417.70 ± 9.08	6473.71 ± 9.09	209.34 ± 9.08	327.88 ± 2.33	859.49 ± 0.09	1534.25 ± 9.80	5.91 ± 0.07	1.51 ± 0.04	118.91 ± 0.98
S7P1	**City Centre 2**	898.00 ± 8.99	342.09 ± 9.88	5783.09 ± 7.88	199.11 ± 8.72	313.09 ± 9.11	745.12 ± 9.87	899.98 ± 7.99	4.55 ± 0.03	0.99 ± 0.01	72.34 ± 1.22
S7P2		134.96 ± 89	59.44 ± 7.23	1655.64 ± 7.89	361.66 ± 8.99	107.84 ± 7.44	115.09 ± 9.11	279.42 ± 12.22	0.75 ± 0.02	1.21 ± 0.09	14.07 ± 2.98
S7P3		639.93 ± 3.99	113.02± 8.11	1400.36 ± 3.45	550.85 ± 7.99	67.60 ± 12.98	106.08 ± 9.88	330.92 ± 8.77	2.84 ± 1.11	0.32 ± 0.04	20.80 ± 7.44
S8P1	**Residential**	103.52 ± 8.99	104.15 ± 2.34	2557.41 ± 7.88	674.07 ± 12.22	120.37 ± 9.09	245.66 ± 9.00	831.13 ± 12.11	2.11 ± 0.22	1.23 ± 0.09	166.01 ± 0.45
S8P2		41.08 ± 0.99	12.05 ± 2.42	210.27 ± 9.87	6.63 ± 0.09	13.13 ± 9.88	72.66 ± 7.77	97.76 ± 6.77	0.33 ± 0.05	0.13 ± 0.01	3.62 ± 0.09
S8P3		196.86 ± 0.99	50.34 ± 9.12	1147.98 ± 7.11	407.53 ± 2.33	65.11 ± 8.08	59.56 ± 9.88	182.44 ± 0.99	1.22 ± 0.02	0.22 ± 0.00	7.67 ± 0.98

* Fe results are presented in mg/g due to the high values recorded from the study sites.

**Table 4 ijerph-19-10238-t004:** Minimum and maximum concentrations (ug/g and **mg/g ***) of the trace metals (as shown in Table 3) in the soil from the study area in comparison with FAO and WHO standards.

Trace Element	Min	Average	Maximum	MRL(Soil) [23]
As	0.09	2.68	8.42	20.0
Pb	1.16	47.06	188.34	100.0
Cd	0.06	0.70	1.59	3.0
Cr	5.80	140.14	417.70	100.0
Ni	8.91	138.86	486.60	50.0
Zn	40.56	556.76	1675.21	-
Mn	144.14	2715.04	9814.90	2000
**Fe ***	**3.06**	**154.54**	**674.07**	**5000**
Cu	10.51	215.21	859.49	100.0
Ti	19.94	357.26	898.0	

* The average values were calculated from all the recorded results for each trace elements as shown in Table 3.

**Table 5 ijerph-19-10238-t005:** Pollution index summary for each of the sites.

Sample	Cr	Cu	Zn	Pb	Ni	Cd
S1P1	0.18	0.27	0.29	0.03	0.25	0.04
S1P2	0.13	0.25	0.35	0.05	0.19	0.15
S1P3	0.09	0.17	0.20	0.02	0.18	0.06
S2P1	1.19	1.50	1.59	0.29	1.44	0.24
S2P2	1.46	1.37	1.44	0.25	1.12	0.19
S2P3	0.18	0.63	0.41	0.06	0.21	0.07
S3P1	1.61	3.48	3.74	0.89	2.06	0.56
S3P2	3.14	3.56	2.96	0.55	3.78	0.58
S3P3	5.66	6.00	4.70	0.95	6.52	0.94
S4P1	2.85	4.39	4.09	1.64	4.54	0.69
S4P2	1.65	1.99	2.02	0.32	2.18	0.30
S4P3	6.30	6.32	5.85	1.06	9.73	1.14
S5P1	1.21	2.37	2.58	0.38	1.36	0.35
S5P2	0.72	1.63	1.57	0.18	0.93	0.19
S5P3	0.54	1.43	0.96	0.23	0.69	0.70
S6P1	3.81	6.42	8.38	2.69	4.49	1.01
S6P2	4.66	5.23	4.90	0.79	6.68	0.80
S6P3	6.53	13.64	7.67	1.70	6.56	1.08
S7P1	5.35	11.83	4.50	1.03	6.26	0.71
S7P2	0.93	1.83	1.40	0.20	2.16	0.86
S7P3	1.77	1.68	1.65	0.30	1.35	0.23
S8P1	1.63	3.90	4.16	2.37	2.41	0.88
S8P2	0.19	1.15	0.49	0.05	0.26	0.09
S8P3	0.79	0.95	0.91	0.11	1.30	0.16
**Average**	**2.19**	**3.42**	**2.78**	**0.67**	**2.78**	**0.50**

**Table 6 ijerph-19-10238-t006:** Geoaccumulation Index.

Sample	Cr	Cu	Zn	Pb	Ni	Cd
S1P1	−5.48	−3.42	−2.36	−6.06	−4.15	−6.23
S1P2	−5.97	−3.53	−2.08	−5.34	−4.58	−4.42
S1P3	−6.50	−4.10	−2.89	−7.01	−4.66	−5.81
S2P1	−2.79	−0.93	0.08	−2.88	−1.65	−3.77
S2P2	−2.49	−1.06	−0.06	−3.08	−2.01	−4.06
S2P3	−5.52	−2.17	−1.88	−5.24	−4.45	−5.49
S3P1	−2.35	0.29	1.32	−1.26	−1.13	−2.53
S3P2	−1.38	0.32	0.98	−1.96	−0.25	−2.47
S3P3	−0.54	1.07	1.65	−1.17	0.54	−1.78
S4P1	−1.53	0.62	1.45	−0.39	0.01	−2.21
S4P2	−2.31	−0.52	0.43	−2.76	−1.05	−3.42
S4P3	−0.38	1.15	1.96	−1.01	1.11	−1.50
S5P1	−2.76	−0.27	0.78	−2.51	−1.72	−3.20
sS5P2	−3.51	−0.81	0.06	−3.58	−2.28	−4.11
S5P3	−3.91	−1.00	−0.64	−3.24	−2.71	−2.20
S6P1	−1.11	1.17	2.48	0.33	0.00	−1.67
S6P2	−0.81	0.87	1.71	−1.45	0.57	−2.01
S6P3	−0.33	2.26	2.35	−0.34	0.54	−1.58
S7P1	−0.62	2.05	1.58	−1.05	0.48	−2.18
S4P2	−3.14	−0.65	−0.10	−3.41	−1.06	−1.89
S7P3	−2.22	−0.76	0.14	−2.85	−1.73	−3.81
S8P1	−2.33	0.45	1.47	0.15	−0.90	−1.87
S8P2	−5.45	−1.31	−1.62	−5.37	−4.10	−5.11
S8P3	−3.38	0.82	−0.72	−4.29	−1.79	−4.35
**Average**	**−2.78**	**−0.39**	**0.25**	**−2.74**	**−1.54**	**−6.23**

**Table 7 ijerph-19-10238-t007:** Exposure assessment to soil pollution via ingestion, inhalation, or dermal contact.

	ADD_ing_		ADD_inh_		ADD_dermal_	
	Adult	Child	Adult	Child	Adult	Child
As	9.19 × 10^−6^	1.71 × 10^−5^	6.76 × 10^−9^	1.20 × 10^−8^	1.40 × 10^−7^	5.39 × 10^−8^
Pb	1.64 × 10^−4^	3.01 × 10^−4^	1.194 × 10^−1^	2.10 × 10^−7^	2.46 × 10^−6^	9.46 × 10^−7^
Cd	2.40 × 10^−6^	4.47 × 10^−6^	1.77 × 10^−9^	3.13 × 10^−9^	3.65 × 10^−8^	1.41 × 10^−8^
Cr(vi)	4.80 × 10^−4^	8.99 × 10^−4^	3.53 × 10^−1^	6.26 × 10^−7^	7.32 × 10^−6^	2.82 × 10^−6^
Ni	4.76 × 10^−4^	8.88 × 10^−4^	3.50 × 10^−1^	6.21 × 10^−7^	7.25 × 10^−6^	2.79 × 10^−6^
Zn	1.91 × 10^−3^	3.56 × 10^−3^	1.40 × 10^−2^	2.49 × 10^−6^	2.91 × 10^−5^	1.12 × 10^−5^
Mn	9.32 × 10^−3^	0.02	6.85 × 10^−2^	1.21 × 10^−5^	1.42 × 10^−4^	5.46 × 10^−5^
Fe	0.53	0.99	3.90 × 10^−4^	6.91 × 10^−4^	8.07 × 10^−3^	3.11 × 10^−3^
Cu	7.38 × 10^−4^	1.37 × 10^−3^	3.43 × 10^−1^	9.62 × 10^−7^	1.12 × 10^−5^	4.32 × 10^−6^
Ti	1.26 × 10^−3^	2.28 × 10^−3^	3.99 × 10^−4^	1.60 × 10^−6^	1.86 × 10^−5^	7.18 × 10^−6^

**Table 8 ijerph-19-10238-t008:** Hazard Quotient.

Metals	HQ_ing_	HQ_ing_	HQ_inh_	HQ_inh_	HQ_dermal_	HQ_dermal_
	Adult	Child	Adult	Child	Adult	Child
As	0.03	0.04	2.25 × 10^23^	4.00 × 10^−4^	4.67 × 10^−4^	1.80 × 10^−4^
Pb	0.05	0.09	3.40 × 10^23^	6 × 10^−5^	6.86 × 10^−4^	2.70 × 10^−4^
Cd	2.40 × 10^−3^	4.47 × 10^−3^	1.77 × 10^22^	3.13 × 10^−6^	3.65 × 10^−5^	1.41 × 10^−5^
Cr	0.16	0.30	3.0 × 10^−3^	2.09 × 10^−4^	2.44 × 10^−3^	9.40 × 10^−4^
Ni	0.02	0.04	1.75 × 10^23^	3.11 × 10^−5^	3.63 × 10^−4^	1.0 × 10^−4^
Zn	6.37 × 10^−3^	0.01	4.67 × 10^22^	8.30 × 10^−6^	9.67 × 10^−5^	3.73 × 10^−5^
Mn	0.14	0.15	4.89 × 10^23^	1.10 × 10^−4^	1.01 × 10^−3^	3.90 × 10^−4^
Fe	0.76	1.41	5.57 × 10^24^	9.87 × 10^−4^	0.01	4.44 × 10^−3^
Cu	0.02	0.03	1.36 × 10^23^	2.41 × 10^−5^	2.80 × 10^−4^	1.08 × 10^−4^
HI	1.17	2.07	6.99 × 10^24^	1.47 × 10^−3^	0.02	6.52 × 10^−3^

**Table 9 ijerph-19-10238-t009:** Risk Index (RI).

Trace Metal	Adult_ADDing, inh, der_	Children_ADDing, inh, der_	RI Adult/Children
As	6.76 × 10^19^	1.72 × 10^−5^	1.01 × 10^21^/2.58 × 10^−5^
Pb	1.19 × 10^21^	3.02 × 10^−4^	1.01 × 10^19^/2.57 × 10^−6^
Cd	1.76 × 10^19^	4.49 × 10^−6^	8.80 × 10^18^/2.24 × 10^−6^
Cr	3.53 × 10^21^	9.02 × 10^−4^	1.75 × 10^21^/4.51 × 10^−4^
Ni	3.53 × 10^21^	8.91 × 10^−4^	-
Zn	1.40 × 10^22^	3.57 × 10^−3^	-
Mn	6.85 × 10^22^	0.02	-
Fe	3.9 × 10^24^	1.00	-
Cu	5.43 × 10^21^	1.37 × 10^−3^

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
