# Peer review of "Concentrations of Pb and Other Associated Elements in Soil Dust 15 Years after the Introduction of Unleaded Fuel and the Human Health Implications in Pretoria, South Africa"

_ijerph, 2022, doi:10.3390/ijerph191610238_

Round 1
Reviewer 1 Report
The authors investigated soil contamination around gas stations in Pretoria, South Africa. Although the title of the manuscript refers to the investigation of lead contamination, the amount of several other metals (toxic and essential (in given amounts) for the living organisms) was also investigated.
A large number of analytical measurements have been performed and the measurements seems to be correct. At the same time, the interpretation of the results is not appropriate, the reported data are not understandable, and the conclusions are not supported.
In the introduction of the manuscript, the antecedents of the measurements are presented, using a number of references, but at the same time, the biological effect of each metal is written in the discussion section, although this also belongs to the introduction.
The presentation of the methods used to evaluate the results is not clear:
1) 2.3. paragraph "The recoveries for all trace metal ranged from 95 % and 103%." How can the result be greater than 100%?
2) 2.4.1. paragraph: The calculation of the “geoaccumulation index” is included, but the description is not clear, and the references (Müller, 1969, Olowoyo, 2015) is not included in the reference list. Thus, it is not clear what values ​​are considered "geochemical background".
(I should mention that giving references is unusual and it is difficult to find the cited literature. I recommend using the standard referencing method!)
Likewise, the calculation of the "Pollution index" is not understandable, the cited literature (Olowoyo, 2015) is not in the reference list. This would be particularly important because the data from their own previous paper (?) is used as a standard.
3) 3. Results and discussion: The data in the tables are not understandable:
Table 2: the unit of measurement is microgram/g and mg/g, but it is not clear which unit of measurement belongs to the each data.
What does Suburb1, 2, etc. mean? Within this, what is the meaning of S1P1, S1P2, S1P3, S2P1, S2P2, S2P3 ... etc. marking. The locations of the samplings should have been given on a much more detailed map.
Table 3: “Min.”, “Average” and “Max.” data are collected, however, it is not clear, where they come from and how was the average calculated? What does "the study area" mean? What unit of measurements belong to the data? The data are also listed in the text, but for Mn and Fe they do not match the data in the table.
Table 4: The data are difficult to interpret, and it would be needed to include the data used for comparison.
Figure 2: What data is represented?
Table 5: The geo-accumulation indexes are listed here, but the marking of the samples cannot be identified, it differs from the marking used in Table 2. In addition, the Pb (3.065), Cu 2.126 and Zn (2.059) mentioned as examples in the text are not consistent with the data in the table.
Previous data used for comparison would also be necessary here.
4) It was established that, based on the measured data, "Though the HQ results indicated no risk for both adults and children, however, the common route of exposure is via ingestion." and ….all HI values ​​were less(er) than 1, indicating there was no non-carcinogenic risk.” The wording is unclear. According to the last sentence, there is therefore a "carcionogenic risk".
These conclusions are not supported due to uncertain background calculations and lack of comparative data.
Author Response
Dear Reviewer,
Thanks so much for your comments.
We have addressed all your concerns and we do appreciate your effort in making it a better manuscript.
Thanks

Reviewer 2 Report
Summary: The authors present an interesting study that assesses risk associated heavy metals in soil a region 15 years after the introduction of unleaded fuel. The primary focus of the study is to investigate Pb levels and concomitant heavy metals in soil and assess health risks based on calculations of geoaccumlation index (Igeo) and pollution index (PI), as well as other indices for health and exposure.
Comments by line:
Lines 10-11: need a transition between the sentences to indicate that trace metals are associated with fuel
Line 17: correct spelling for “titanium”
Line 18: The concentration ranges should be defined for each metal listed and not listed as a broad range (e.g., Pb = xx-xx ug/g; Fe = xx-xx ug/g; … )
Line 20: It is not surprising that Fe is highest due to its natural occurrence—is this sentence necessary?
Line 22: briefly define pollution index and geo-accumulation index in the abstract
Line 25: clearly list which metal is associated with a carcinogenic vs. non-carinogenic risk.
Line 28: the sentence “The concentrations for Pb were…” needs context. How exactly did Pb compare, and what are the implications?
Line 37: Fe occurs naturally in high abundances and is not typically associated with a risk. Perhaps break this sentence up to list the toxicant metals and the micro-nutrient metals
Line 43: The sentence “… are exceedingly poisonous event at low concentrations” needs an updated reference. Swap term “poisonous” for toxic. Also, toxicity related to low concentration is unlikely unless it is chronic exposure—this should be clarified.
Line 47: replace word “prohibition” with “prohibited usage”. Readers in the US often associate prohibition with the alcohol ban.
Line 47: “… into place decades ago.” List the specific year.
Line 48: The chemical abbreviation for lead (Pb) is defined on line 36, and is continuously reintroduced throughout the manuscript. Please change any subsequent instances of “lead (Pb)…” to just “Pb”.
Line 51: Change the term “toxin” to “toxicant”. Pb is a toxicant, not a toxin (which implies biologic origin).
Line 52: Change “are” to “is” in sentence “… because Pb are non-biodegradable”
Line 53: Change the terms “their blueprints” to “it’s fingerprint”
Lines 65-66: This sentence is unclear and should be rephrased for clarity. Remove redundant language “in some previously published investigations”
Line 72: remove “,” after Pb.
Lines 72-72: needs reference for the statement
Line: 74: needs reference for the statement
Line 82: change “it’s” to “its”
Lines 84-85: Confusing wording—consider breaking into two sentences and rewording for clarity.
Lines 88-89: “… there are few data on the impact…” change “data” to “studies”
Line 94: the health risks associated with Mn should be described and cited accordingly.
Line 98: This needs some context for what the levels mean--how do the Mn and Pb levels compare to national standards (or compare to US EPA maximum contamination levels)? Higher or lower?
Line 98: Round to whole number
Lines 102-105: it is unclear if the goal is to assess the reduction in Pb by comparing it to previous study in the area, or to assess the persistence of legacy Pb contamination following the ban on leaded fuel in 2006. This should be clearly stated.
Line 105: remove additional space between “other_ trace”
Line 130: it sounds like street dust samples were collected as opposed to soil samples. Are these areas where a person might be exposed? Are there any residential homes, schools, or parks surrounding the petrol stations?
Figure 1: low resolution image—consider remaking figure at a higher resolution. Also consider a zoomed-in image to show the sampling areas.
Line 139: clarify if a No. 100 (150 um) mesh was used or a 100 um mesh size.
Line 140: a brief description of the extraction protocol should be listed. Were these leachates or complete digestions?
Lines 137-150: additional details on samples preparation and analytical measurements should be included
Line 154: the purpose of the geoaccuulation index should be explained
Line 158: please describe how the geochemical background values are calculated and list for each metal.
Line 159: how is the background matrix factor determined? Is it the same for all metals? This should be clearly stated in the text.
Line 170: similarly, how is the background value determined? This should be clearly stated in the text and values should be listed for each metal.
Line 175: it is unclear what is meant by “greater than unity”. Please rephrase for clarity.
Lines 177-181: a more detailed explanation for evaluating human health risk is needed.
Lines 186-189: define all terms in the equations
Table 1: define the units of the exposure values and include a brief description of what is being shown.
Line 211: are there relations whips between the trace metals, and if so, they should be described or listed in a table
Table 2: list the units next to each metal in the column heading (e.g., ug/g or ng/g); reduce the decimal place to whole numbers to accurately reflect the precision of the measurement.
Line 218: briefly explain why Pretoria city centers might be higher compared to other sites
Lines 223-228: This needs some context for what the levels mean--how do the levels compare to national standards (or compare to US EPA maximum contamination levels)? Higher or lower?
Line 235-241: Fe, Mn and Zn are micronutrients with both geologic and environmental sources—additional literature search is recommended to aide in the discussion of these metals.
Table 3: list the units of measurement. Reduce the number of decimal places to a whole number for values >10. Reduce the decimal places accordingly to accurately reflect the true analytical precision. List the detection limits in the caption.
Line 243: describe the trend and compare values
Line 264: remove or rephrase “with the majority of its compounds being toxic in nature”. Clarify whether you mean that Pb-based compounds are toxicants or all isotopes of Pb are toxicants.
Lines 269-288: The sources and health risks of Pb and Zn would be better suited in the introduction.
Lines 297-300: citations needed to support discussion on sources of Mn.
Line 300: change “tyres” to “tires”
Table 4: please include information on sample locations (list in order of location) and define the pollution index value in the caption.
Line 346: continental crust represents a broad range of geologic rock types and is not appropriate description of the source of fuel. Rephrase sentence to remove continental crust, or clarify.
Figure 2: define pollution index in caption. It is still unclear what is the purpose of using pollution index value, and should be better explained
Table 5: briefly define geo-accumulation index in caption
Table 6: what are the units on these values? Please explain the relationship between the value (low vs high) and the risk in the discussion.
Line 380: could you expand on findings here? Does dermal contact refer to absorption through the skin? My understanding is that inhalation poses a greater risk for exposure than dermal absorption. However, dermal contact followed by hand-to-mouth behavior can introduce contaminants—would this be considered ingestion? Some clarification and explanation would be helpful here.
Line 409: The risk of Fe and Mn are not previously discussed in the context of the RI calculations, making this seem irrelevant. In addition, the presence of ferric vs. ferrous Fe will influence exposure risk based on the differences in solubility of the oxidation states.
Line 420: change “lesser” to “less”
Reference information needed:
Line 435: missing volume and page numbers
Line 453: missing title
Line 457: missing information and date
Line 463: provide a link to the university library or doi for dissertation
Line 480: missing volume and page numbers
Line 489: missing volume
Line 514: missing volume
Line 516: missing volume and page numbers
General Comments and recommendation:
The manuscript would benefit from a more detailed explanation of human health risk in relation to the metals discussed, particularly micronutrients Fe and Mn which are highlighted as significant in both the introduction and conclusion. The use of calculated risk assessment was unclear in the early portions of the discussion, without some context relating to the maximum contamination levels. In other words, the widespread presence does not necessarily indicate a risk, but a comparison on whether the metals levels exceed some national standards would make this more clear and convincing. This was addressed later, but made the early sections of discussion confusing. In addition, the authors are encouraged to conduct proof-reading for reader clarity, a thorough review of concentration and index values presented and the appropriate units, and a more thorough explanation of health risks as they relate to the values of HQ, RI, HI, PI, and Igeo. The main findings are that children are most at risk through ingestion, however it is unclear if children are exposed in this area and are truly at risk. The manuscript would benefit from a more in-depth discussion of the at-risk population and how individuals might be exposed to this area, if at all. The main take-away is that there are persistent heavy metals present in areas surrounding petrol stations following the banning of leaded-gasoline, and that the continued levels may be a combination of both legacy contamination and traffic/combustion debris, although not explicitly stated.
Author Response
Dear Reviewer
Thanks so much for your detailed revised work.
Highly appreciated. This has certainly improve the quality of out manuscript.
Thanks

Round 2
Reviewer 1 Report
I accept most of the answers and revision of the manuscript. However, I still reserve some of my comments.
1) The authors investigated soil contamination around gas stations in Pretoria, South Africa. Although the title of the manuscript refers to the investigation of lead contamination, the amount of several other metals (toxic and essential (in given amounts) for the living organisms) was also investigated.
I still recommend changing the title of manuscript.
2) Table 3: “Min.”, “Average” and “Max.” data are collected, however, it is not clear, where they come from and how was the average calculated? The title of Table 3 should describe that the minimum and maximum are the lowest and highest values ​​measured and reported in Table 2, while the average values are the average of all data for each metal in Table 2.
3) Page 7, row 524: „indicating there was no non-carcinogenic risk”
The wording is unclear. I suggest rewording the sentence.
Author Response
Dear Editor,
Thanks so much for the assistance provided.
Thanks
